Endemism and diversity of small mammals along two neighboring Bornean mountains

http://orcid.org/0000-0002-6385-7963 Camacho-Sanchez Miguel 1 2 miguelcamachosanchez@gmail.com
Hawkins Melissa T.R. 3 4 5 Melissa.hawkins@humboldt.edu
Tuh Yit Yu Fred 6
http://orcid.org/0000-0002-4282-1072 Maldonado Jesus E. 3
http://orcid.org/0000-0003-0291-7819 Leonard Jennifer A. 1
1 Conservation and Evolutionary Genetics Group, Doñana Biological Station (EBD-CSIC) , Sevilla , Spain
2 CiBIO—Centro de Investigação em Biodiversidade e Recursos Genéticos da Universidade do Porto , Vairão , Portugal
3 Center for Conservation Genomics, Smithsonian Conservation Biology Institute, National Zoological Park , Washington, DC , USA
4 Department of Biological Sciences, Humboldt State University , Arcata, CA , USA
5 Division of Mammals, National Museum of Natural History , Washington, DC , USA
6 Sabah Parks , Kota Kinabalu, Sabah , Malaysia
Garant Dany
Electronic publication date: 2019 Oct 8
Publication date: 2019
Volume: 7
Electronic Location ID: e7858
Received 2018 Feb 14; Accepted 2019 Sep 9
Copyright: © 2019 Camacho-Sanchez et al.
Copyright year: 2019
Copyright holder: Camacho-Sanchez et al.
License: This is an open access article distributed under the terms of the Creative Commons Attribution License, which permits unrestricted use, distribution, reproduction and adaptation in any medium and for any purpose provided that it is properly attributed. For attribution, the original author(s), title, publication source (PeerJ) and either DOI or URL of the article must be cited.
License URL: https://creativecommons.org/licenses/by/4.0/

Keywords: Mt. Kinabalu, Mt. Tambuyukon, Shannon index, Elevational gradient, Sundaland

Funding: Spanish Ministry of Science and Innovation Predoctoral Fellowship BES-2011-049186 and EEBB-I-12-05317 Spanish Research Council CGL2010-21524, CGL2014-58793-P Miguel Camacho-Sanchez was supported by the Spanish Ministry of Science and Innovation Predoctoral Fellowship BES-2011-049186 and his fieldwork in Borneo was supported by grant EEBB-I-12-05317. This research was supported by the Spanish Research Council grants CGL2010-21524, CGL2014-58793-P. The funders had no role in study design, data collection and analysis, decision to publish, or preparation of the manuscript.

==============================
Mountains offer replicated units with large biotic and abiotic gradients in a reduced spatial scale. This transforms them into well-suited scenarios to evaluate biogeographic theories. Mountain biogeography is a hot topic of research and many theories have been proposed to describe the changes in biodiversity with elevation. Geometric constraints, which predict the highest diversity to occur in mid-elevations, have been a focal part of this discussion. Despite this, there is no general theory to explain these patterns, probably because of the interaction among different predictors with the local effects of historical factors. We characterize the diversity of small non-volant mammals across the elevational gradient on Mount (Mt.) Kinabalu (4,095 m) and Mt. Tambuyukon (2,579 m), two neighboring mountains in Borneo, Malaysia. We documented a decrease in species richness with elevation which deviates from expectations of the geometric constraints and suggests that spatial factors (e.g., larger diversity in larger areas) are important. The lowland small mammal community was replaced in higher elevations (from above ~1,900 m) with montane communities consisting mainly of high elevation Borneo endemics. The positive correlation we find between elevation and endemism is concordant with a hypothesis that predicts higher endemism with topographical isolation. This supports lineage history and geographic history could be important drivers of species diversity in this region.

Introduction

Understanding the mechanisms that are responsible for shapping patterns of biodiversity across geography has been an important driver of biological research (Wallace, 1869; Heaney, 1986; Rosenzweig, 1995; Lomolino et al., 2010). Mountains are valuable natural experiments that allow researchers to test biogeoraphical hypotheses for these reasons: (1) they have limited confounding variation across historical and ecological conditions, (2) they are discrete units to study and (3) they offer replicated gradients for factors (climatic, spatial, ecological) that have been central for research in biogeography (Brown, 2001). Numerous studies have sought to explain the change of alpha diversity across elevation for diverse taxonomic groups (Rahbek, 1995; Patterson et al., 1998; Heaney, 2001; Fu et al., 2006; Kluge, Kessler & Dunn, 2006). Previous research on diversity gradients on mountains has focused on the relationship between diversity gradients on mountains with temperature and precipitation, primary productivity, area, isolation, and geometric constraints. However, the effects of abiotic predictors on mountains are inconsistent across studies (Rahbek, 1995; Patterson et al., 1998; Heaney, 2001; Fu et al., 2006; Kluge, Kessler & Dunn, 2006), and to date, no uniform theory explains mammalian diversity gradients on mountains (Brown, 2001; Lomolino, 2001; Heaney, 2001; Stevens, Rowe & Badgley, 2019).

Geometric constraints stand out from the former predictors as the most recurrent explanation for diversity gradients on mountains across many systems (Rahbek, 1995; Fu et al., 2006; Kluge, Kessler & Dunn, 2006; Rowe, 2009). They predict that the overlap of species’ ranges on a constrained area causes species richness to be higher at the center of this area (Colwell & Lees, 2000). Translated to mountains, geometric constraints predict a higher diversity at mid-elevations (mid-elevation bulge) caused by the overlap of species ranges with midpoints at different elevations. This phenomenon is often referred to as mid-domain effect (MDE). In mammals, a MDE has been reported across different mountain ranges, but it is not a global pattern (Rickart, 2001; Heaney, 2001; Nor, 2001; Li, Song & Zeng, 2003; McCain, 2004, 2005; Rowe, 2009; Rowe, Heaney & Rickart, 2015; Hu et al., 2017).

The history of lineage and place can profoundly influence current patterns of diversity on mountains (Brown, 2001; Lomolino, 2001; Stevens, Rowe & Badgley, 2019). These effects of history vary from mountain system and taxa, which has hampered the consolidation of a theory to explain species richness on mountain gradients (Brown, 2001). On mountaintops, (1) more geographical isolation should favor speciation and restrict processes of dispersal and colonization, and (2) species richness should decline as a consequence of higher extinction caused by generally smaller areas associated with higher elevation bands (Brown, 2001; Lomolino, 2001). This, in turn, should lead to fewer species but higher endemism on mountaintops. A recent worldwide review on plant endemism on mountains by Steinbauer et al. (2016) found a clear correlation of peak isolation with increased endemism in mountaintops. They suggested the coupling of historical climate changes with topography as a “pump” for montane endemism (Steinbauer et al., 2016). For mammals, there is growing evidence that endemism increases with elevation (Sánchez-Cordero, 2001; Heaney, 2001; Swenson et al., 2012; Chen et al., 2017), but this data is often descriptive and lacks statistical testing.

Here, we surveyed non-volant small mammals across elevational gradients on two tropical mountains from the same range, Mt. Kinabalu and Mt. Tambuyukon, in northern Borneo. Mt. Kinabalu (4,095 m) is the tallest mountain in Sundaland, a tropical hotspot of biodiversity (Myers et al., 2000). A number of taxa have been surveyed across large elevational transects on this mountain: moths (Beck & Chey, 2008), ants (Brühl, Mohamed & Linsenmair, 1999; Malsch et al., 2008), plants (Kitayama, 1992; Aiba & Kitayama, 1999; Aiba, Takyu & Kitayama, 2005; Grytnes & Beaman, 2006; Grytnes et al., 2008), oribatid mites (Hasegawa, Ito & Kitayama, 2006), snails (Liew, Schilthuizen & Bin Lakim, 2010) and small mammals (Nor, 2001). These studies have recovered either a decline in diversity with elevation, which seems to fit a global pattern (Rahbek, 1995) or a MDE. To date, Nor’s (2001) survey is the most comprehensive dataset of non-volant small mammals on Mt. Kinabalu. It describes a clear MDE, although the number of species he reports in the lowest elevation was unexpectedly low (only five species), compared to the maximum of approximately 35 non-volant small mammals that are known to be distributed in the lowland forests of Borneo (Nor, 2001; Payne et al., 2007). Thus, there could be a strong bias in the interpretations caused by incomplete sampling of the lowest elevation (Rahbek, 1995; Lomolino, 2001). Furthermore, Mt. Kinabalu is a hotspot of endemism for many mountain lineages (Payne et al., 2007; Van Der Ent, 2013; Merckx et al., 2015). The montane conditions in the upper slopes of Mt. Kinabalu are unique in Sundaland. Similar conditions are only partially found on a handful of other peaks in Borneo (i.e., upper montane forest in Mt. Tambuyukon, 2,579 m, Wood & Van Der Ent, 2012; Mt. Trusmadi, 2,642 m, Kitayama et al., 1993; Mt. Murud, 2,423 m, Beaman & Anderson, 1997; and Mt. Mulu, 2,376 m, Collins, 1980) and on several peaks in Sumatra (Laumonier, 1997).

This study aimed to describe the effects of (1) the MDE on alpha diversity (species richness) and (2) mountain endemics on beta diversity (change in the composition of the community), for small non-volant mammals along the elevational gradient on Mt. Kinabalu and Mt. Tambuyukon. We hypothesized that alpha diversity should decrease with elevation coinciding with less available area and less complex habitat heterogeneity, while greater endemism on higher elevational bands should correlate with the particular isolation of Mt. Kinabalu in Sundaland. For this purpose, we characterized the small mammal diversity along the elevational gradient on Mt. Kinabalu and Mt. Tambuyukon. The raw data is fully accessible and all analyses and figures are reproducible (DOI 10.5281/zenodo.3341178).

Materials and Methods

Study sites

Mt. Kinabalu and Mt. Tambuyukon are two neighboring peaks inside Kinabalu National Park in the Malaysian state of Sabah, Borneo (Fig. 1). This park covers 764 square kilometers. Mt. Kinabalu is the tallest peak in Borneo at 4,095 m and is home to thousands of endemic plant and animal species (Payne et al., 2007; Van Der Ent, 2013). Mt. Tambuyukon (the third highest peak in Borneo, 2,579 m; Fig. 1), despite being only 18 km away, is far less scientifically explored. The vegetation zones as described by Kitayama (1992) for Mt. Kinabalu have been used for simplicity as well as for consistency with previous elevational surveys (Nor, 2001): lowland (>1,200 m), lower montane (1,200–2,000 m), upper montane (2,000–2,800 m) and subalpine (2,800–3,400 m).

Figure 1 Trapping locations.

Map of Kinabalu Park, Sabah, Malaysia with trails followed and trapping locations.

The lowland dipterocarp forest dominates both mountains from the lowest elevations up to 1,200 m. Above this elevation begins the lower montane oak forest of 10-25 m trees up to around 1,800-1,900 m on both Mt. Kinabalu and Mt. Tambuyukon. On Mt. Tambuyukon at 1,440 m there is a sharp break to an ultramafic outcrop and the vegetation changes to a low productivity forest with shorter trees. The mossy or cloud forest begins at around 2,000 m on both mountains. This zone is usually immersed in clouds, moss covers most surfaces, and pitcher plants (genus Nepenthes), epiphytes, orchids, and climbing bamboos are abundant. At 2,350 m on Mt. Tambuyukon and 2,600 m on Mt. Kinabalu there is a fast transition to an open stunted forest dominated by Dacrydium and Leptospermum species. At these elevations, the vegetation develops a sclerophyllous and microphyllous syndrome. At 2,800 m the subalpine vegetation appears on Mt. Kinabalu, which is absent on Mt. Tambuyukon.

Field survey

Surveys were conducted in two consecutive field seasons along elevational gradients following climbing trails along Mt. Tambuyukon and Mt. Kinabalu. We targeted small non-volant mammals and further included opportunistic observations and data from trail cameras. Species identification was performed according to Payne et al. (2007). During the first field season, we surveyed Mt. Tambuyukon in June–August 2012. Surveys for the second field season were conducted on select locations on Mt. Tambuyukon (to fill gaps in the first season sampling) and along the full elevational gradient of Mt. Kinabalu in February–April 2013.

We set traps from ~331 to 2,509 m on Mt. Tambuyukon, and from 503 to 3,466 m on Mt. Kinabalu (Dataset S1). The taxa we expected in the small mammal trap surveys included members of the families Soricidae (shrews), Erinaceidae (gymnures), Tupaiidae (treeshrews), and rodents in the families Muridae (mice and rats) and Sciuridae (squirrels). Trapping was conducted following ethical standards according to the guidelines of the American Society of Mammalogists (Sikes, Gannon & Animal Care and Use Committee of the American Society of Mammalogists, 2011). Animal care and use committees approved the protocols (Smithsonian Institution, National Museum of Natural History, Proposal Number 2012-04 and Estación Biológica de Doñana Proposal Number CGL2010-21524). Field research was approved by Sabah Parks (TS/PTD/5/4 Jld. 45 (33) and TS/PTD/5/4 Jld. 47 (25)), the Economic Planning Unit (100-24/1/299) and the Sabah Biodiversity Council (JKM/MBS.1000-2/2(104)).

Line transects were set at approximately every 400–600 m in elevation. On Mt. Tambuyukon, transects were placed along the mountaineering trail markers (placed every one km along the trail) as follows: from Monggis substation to km 1 at 500 m, 900 m (km 7.5), 1,300 m (km 10.3), 1,600 m (km 11), 2,000 m (km 12.6) and 2,400 m (km 13.5). On Mt. Kinabalu the 500 and 900 m transects were located at Poring Hot Springs, from the entrance and along the trail to the Langanan Waterfall. The next elevation transect for Mt. Kinabalu was set at ~1,500 m at the Park Headquarters, ~2,200 m along the Timpohon mountaineering trail (km 2, Kamborangoh), 2,700 m (km 4, Layang-Layang), and 3,200 m (around Waras, Pendant hut and Panar Laban). For reporting results and data analysis, we hereafter refer to these locations as “trapping locations.” Each trapping location gathered the trap data from transects which could span a distance of over 100 m up- or downhill (raw data in Dataset S1).

We set traps at approximately 5–10 m intervals for a total of around 40 traps per transect. Trapping locations are shown in Fig. 1. Collapsible Tomahawk live traps (40 cm long), collapsible Sherman traps (two sizes 30 and 37 cm long), and local mesh-wire box traps were used. We considered traps as “close to the ground” if set below approximately three meters off the ground. Most of these traps were directly set on the ground, while a small proportion was set on branches or vines at the reach of the hand. We considered the three-meter threshold as reasonable to describe the terrestrial small mammal community given the arboreal habits of many small terrestrial Bornean mammals (Wells et al., 2004) and the high-complexity of the vegetation in most transects. Any trap above that Threshold was considered “arboreal.” A bait mixture (of varying composition) consisting of bananas, coconuts, sweet potatoes, palm fruit and oil, vanilla extract and dried fish was placed in each trap. A small number of pitfall traps were distributed from 500 to 2,000 m on Mt. Tambuyukon (number and distribution were limited by the substrate) (Table 1).

Table 1 Trap success across all elevations.

The number of animals caught is in column N, followed by number of trap nights, and the overall trap success per trapping location.

		Including arboreal and pitfall traps	Excluding arboreal and pitfall traps	
Elev. (m)	N	Trap nights	Trap success (%)	N	Trap nights	Trap success (%)	
Mt. Kinabalu	500	33	300	11.0	30	285	10.5	
900	20	360	5.6	20	360	5.6	
1,500	36	360	10.0	36	360	10.0	
2,200	35	434	8.1	34	427	8.0	
2,700	60	390	15.4	60	390	15.4	
3,200	29	200	14.5	29	200	14.5	
Totals	213	2,044	10.4	209	2,022	10.3	
Mt. Tambuyukon	500	78	1,713	4.6	75	1,588	4.7	
900	24	992	2.4	24	956	2.5	
1,300	53	712	7.4	52	702	7.4	
1,600	22	1,036	2.1	22	1,025	2.1	
2,000	55	1,036	5.3	55	988	5.6	
2,400	67	698	9.6	67	698	9.6	
Totals	299	6,187	4.8	295	5,957	5.0	

Each trapping location had a total of two to four transects. The highest elevation had a lower number of trap nights due to the smaller area available for placement of traps. Coordinates for trapping locations were recorded using Garmin eTrex® series and Garmin GPSmap 60CSx. The minimum number of trap nights was based on the saturation rates obtained from Nor (2001) at approximately 300 trap nights. For every trapping location the cumulative trap-nights and species richness were calculated by adding the cumulative number of species caught and trap-nights for each trap for each successive day from the date the trap was set.

We set up four camera traps (Reconyx RapidFire RC55™ cameras, and ScoutGuard HCO™ cameras) along the mountaineering trail on Mt. Tambuyukon. Camera 1 was placed at 500 m, at the first-kilometer marker for the hiking trail. Cameras 2 and 3 were placed along the Kepuakan River near km 8 and at approximately 900 m. Camera 4 was placed at approximately 1,300 m near km 10.5. No cameras were deployed along the Mt. Kinabalu trail due to a large number of day hikers and mountain climbers.

Additionally, while on Mt. Tambuyukon we opportunistically recorded mammal observations while walking to, from and along our trap lines, while setting cameras, or while in our campsite.

Alpha diversity

Diversity index

We computed diversity indices for each trapping location. The Community Ecology Package “vegan” 2.5-5 (Dixon, 2003) in R 3.5.1 (R Development Core Team, 2018) was used to calculate the Shannon diversity (H′) and Simpson’s diversity (D). Pielou’s evenness (J′) was calculated as J′ = H′/Hmax, and species richness (S) as the number of species. We used the LOWESS smoother (stats::lowess function, in R) to visualize the change of these indexes with elevation. The pattern of species richness across elevation in each mountain was contrasted against a MDE using “rangemodelR” 1.0.4 (Marathe, 2019) in R, as in Wang & Fang (2012). We used the function range.shuffle which returns the pattern of species diversity under the MDE. The following arguments were used for computations: 50 m as the interval to discretize species ranges and midpoints into elevational bands, soft boundaries and 5,000 repetitions.

Predictors of species richness

The relation between species richness and two predictors, (1) elevation and (2) expectations under the MDE, were assessed in separate linear models with mixed-effects (Laird & Ware, 1982) using lme function from “nlme” package 3.1-137 (Pinheiro et al., 2019) in R. Model building and the evaluation of their fit was carried out following the recommendations in Harrison et al. (2018). In both models, we corrected for spatial autocorrelation by including a term to account for the correlation structure built with nlme::corSpatial function in R. This object contained the centroids of coordinates from all animals trapped at a given location. Mountain was included as a random intercept. The fit of these models was compared with a likelihood ratio tests (King, 1998) against a null model that excluded the fixed effect (Nickerson, 2000; Pinheiro & Bates, 2000; Harrison et al., 2018).

Endemism

We quantified endemism at each trapping location as the proportion of endemic species. This measure is robust to sampling bias and it is not overridden by local diversity (Steinbauer et al., 2016). We defined endemic as only found in Borneo. Shrews were excluded as we were not able to identify them to the species level. To visualize the change of proportion of endemics with elevation we created a confidence envelope by resampling 1,000 times the species present at each trapping location. This same approach was used to visualize the proportion of captures that belonged to endemic species across elevation. We evaluated the contribution of elevation (predictor) to explaining the proportion of endemism (dependent variable 1) and the proportion of captures (dependent variable 2) that corresponded to endemic species. We used generalized models with mixed effects using a binomial family with logit link. Mountain was added as a random intercept. Model fitting was done using glmer function from “lme4” package 1.1-21 (Bates et al., 2015) in R. The fit of these models was compared using likelihood ratio tests against a null model that excluded the fixed effect (Nickerson, 2000; Pinheiro & Bates, 2000).

Beta diversity

We calculated beta diversity for each mountain using a Sorensen-based dissimilarity index (βSOR) with its turnover (βSIM) and nestedness (βNES) components (Baselga, 2010). For calculations we used function beta.multi, from package “betapart” 1.3 (Baselga & Orme, 2012) in R. These calculations were also done pairwise between all trapping locations within the same mountain using the function betapart::beta.pair. These distances were used for a cluster analysis using neighbor-joining (Saitou & Nei, 1987) with ape::nj function in R to evaluate the community similarity between mountains and elevations.

We estimated the contribution of endemics to βSIM and βNES components of beta diversity. For each mountain, we removed endemic species from the dataset and recalculated βSIM and βNES (hereafter referred to as βSIM-end and βNES-end). We created subsets of the data to which these measures could be directly compared by randomly sub-sampling 5,000 times without replacement, several species from the complete dataset equal to the non-endemics present in that mountain. P-values were determined from the distributions of the permutated values.

Results

Field survey

The overall trapping success increased toward high elevation on both mountains. On Mt. Tambuyukon, we trapped a total of 295 different individuals (not including recaptured animals) from 21 different species (Dataset S1; Table S1) over 5,957 trap nights, for a total of 5.0% trap success (not including arboreal or pitfall trapping; Table 1). Trap success at each elevation ranged from 2.1% at 1,600 m to 9.6% at 2,400 m. One non-target capture of a carnivore, a Kinabalu ferret-badger (Melogale everetti), was recorded. The trap success calculations were done excluding pitfall traps and arboreal traps (due to inconsistent placement of traps). The use of pitfall and arboreal traps was limited by the time necessary to set and check arboreal traps, difficulty finding appropriate places to set pitfall traps and the high number of tourists on Mt. Kinabalu, which limited the sites we could set traps to those out of sight of the trails. The accumulation of species across trap nights varied across elevations, and appeared near saturation in all trapping locations (Fig. S1). A species of shrew, Suncus sp., was collected in a pitfall trap, and a gray tree rat (Lenothrix canus) in an arboreal trap, bringing the total number of species to 23.

On Mt. Kinabalu, we trapped a total of 20 species from 209 different individuals (Dataset S1; Table S1) over 2,022 trap nights, for an average trap success of 10.3% (Table 1). The trap success across elevations was much higher on Mt. Kinabalu, ranging from 5.6% (at 900 m), to 15.4% (at 2,700 m) (Table 1). This overall higher capture rate resulted in species saturation with a lower number of trap nights on Mt. Kinabalu than on Mt. Tambuyukon (Fig. S1).

Species distribution

The mountain treeshrew, Tupaia montana, was the most frequently caught species (35.7% of all catches) and it had a wide elevational distribution from 836 to 3,382 m. The Bornean mountain ground squirrel (Sundasciurus everetti, formerly Dremomys everetti; Hawkins et al., 2016), the long-tailed giant rat (Leopoldamys sabanus), and Whitehead’s spiny rat (Maxomys whiteheadi) also had large elevational distributions on both mountains (Fig. 2). The lowland (<1,000 m) terrestrial small mammal community was the most diverse, with 19 species trapped. We captured 16 species in the community associated with montane forest between 1,000 and 2,400 m and only seven species at 2,400 m and above (Fig. 2).

Figure 2 Species distribution across elevations.

Our two field surveys are represented by circles (Mt. Kinabalu), and triangles (Mt. Tambuyukon), together with a previous small mammal survey on Mt. Kinabalu (open squares; Nor, 2001). Bornean endemics are bolded. We have represented the vegetation levels as described in Kitayama (1992) for Mt. Kinabalu in grayscale in the background.

We captured a single shrew (Suncus sp.) after 176 pitfall trap nights. A less intensive arboreal trapping effort of 76 trap nights yielded seven individuals from six small mammal species: T. montana (n = 2), Lenothrix canus (n = 1), Callosciurus prevostii (n = 1), Sundasciurus jentinki (n = 1), Sundamys muelleri (n = 1) and T. minor (n = 1). Despite the smaller effort on arboreal trap nights, we still captured two species that were not trapped elsewhere (Lenothrix canus and Sundasciurus jentinki). However, we also caught arboreal species in ground traps or traps close to the ground (<3 m), including Chiropodomys pusillus, Callosciurus prevostii and T. minor.

We set four trail cameras on Mt. Tambuyukon to document larger mammals not targeted by our traps. They documented an additional eight species of mammals (Table 2; Fig. S2). The number of species captured by the cameras varied from one to five, with the camera at 500 m exhibiting the most diversity, both in number of species and number of independent visits (Table 2). They documented four species that were not documented in any other way.

Table 2 Results of camera trap survey.

Results of camera trap surveys on Mt. Tambuyukon, with relative abundance calculated for 100 trap nights.

Camera	Elevation (m)	Common name	Species	No. of series	Camera nights	Relative abundance	
1	500	Pig-tailed Macaque	Macaca nemestrina	1	42	2.38	
Common Porcupine	Hystrix brachyura	2		4.76	
Mouse Deer	Tragulus sp.	2		4.76	
Muntjac	Muntiacus sp.	1		2.38	
Sambar Deer	Rusa unicolor	1		2.38	
2	900	Malay Civet	Viverra tangalunga	2	42	9.52	
Banded Linsang	Prionodon linsang	1		2.38	
3	900	Malay Civet	Viverra tangalunga	2		–	
4	1,300	Malay Civet	Viverra tangalunga	1	29	3.45	
Masked Palm Civet	Paguma larvata	1		3.45	

On Mt. Tambuyukon, several species were detected only through direct observation (Table 3). Of these sightings many were documented only a single time, including the orangutan (Pongo pygmaeus), the Bornean giant tufted ground squirrel (Reithrosciurus macrotis), Whitehead’s squirrel (Exilisciurus whiteheadii), and the bearded pig (Sus barbatus). The Bornean gibbon (Hylobates muelleri) was heard singing on an almost daily basis, but only directly observed a single time. The sambar deer (Rusa unicolor) was heard vocalizing once at 1,400 m. Only one observation was made of a carnivore, the Malay civet (Viverra tangalunga), which was observed during a late night-walk. The visual observations increased the diversity of species documented, especially for primates and tree squirrels.

Table 3 All species recorded on Mt. Tambuyukon.

Number	Family	Common name	Scientific name	Method(s) of detection	
1	Cercopithecidae	Pig-tailed Macaque	Macaca nemestrina	Camera trap/observation	
2	Cercopithecidae	Long tailed Macaque	Macaca fascicularis	Observation	
3	Cercopithecidae	Maroon Langur	Presbytis rubicunda	Observation	
4	Cervidae	Muntjac	Muntiacus sp.	Camera trap	
5	Cervidae	Sambar Deer	Cervus unicolor	Camera trap/observation	
6	Erinaceidae	Lesser Gymnure	Hylomys suillus	Live trap	
7	Hylobatidae	Bornean Gibbon	Hylobates muelleri	Observation	
8	Hystricidae	Common Porcupine	Hystrix brachyura	Camera trap	
9	Muridae	Common Pencil-tailed Tree Mouse	Chiropodomys pusillus	Live trap	
10	Muridae	Grey tree rat/Sundaic Lenothrix	Lenothrix canus	Live trap	
11	Muridae	Long-tailed giant rat	Leopoldomys sabanus	Live trap	
12	Muridae	Bornean Mountain Maxomys	Maxomys alticola	Live trap	
13	Muridae	Chestnut-bellied spiny rat	Maxomys ochraceiventer	Live trap	
14	Muridae	Brown Spiny Rat	Maxomys rajah	Live trap	
15	Muridae	Red Spiny Rat	Maxomys surifer	Live trap	
16	Muridae	Whitehead’s Rat	Maxomys whiteheadi	Live trap	
18	Muridae	Dark-tailed tree rat	Niviventer cremrioventer	Live trap	
19	Muridae	Mountain long tailed rat	Niviventer rapit	Live trap	
20	Muridae	Summit Rat	Rattus baluensis	Live trap	
21	Muridae	Polynesian/Pacific rat	Rattus exulans	Live trap	
22	Muridae	Giant Mountain Rat	Sundamys infraluteus	Live trap	
23	Muridae	Muller’s Rat/Sundamys	Sundamys muelleri	Live trap	
24	Mustelidae	Kinabalu ferret-badger	Melogale everetti	Live trap	
25	Pongidae	Bornean Orangutan	Pongo pygmaeus	Observation	
26	Sciuridae	Bornean Mountain Ground Squirrel	Sundasciurus everetti	Live trap	
27	Sciuridae	Low’s squirrel	Sundasciurus lowii	Live trap	
28	Sciuridae	Plantain Squirrel	Callosciurus notatus	Observation	
29	Sciuridae	Kinabalu Squirrel	Callosciurus baluensis	Observation	
30	Sciuridae	Giant Squirrel	Ratufa affinis	Observation	
31	Sciuridae	Jentink’s Squirrel	Sundasciurus jentinki	Live trap	
32	Sciuridae	Whitehead’s Squirrel	Exilisciurus whiteheadi	Observation	
33	Sciuridae	Giant Bornean Tufted Ground Squirrel	Reithrosciurus macrotis	Observation	
34	Soricidae	Shrew	Crocidura sp.	Live trap	
35	Soricidae	Shrew	Suncus sp.	Live trap	
36	Suidae	Bearded Pig	Sus barbatus	Observation	
37	Tragulidae	Mouse Deer	Tragulus sp.	Camera trap	
38	Tupaiidae	Common treeshrew	Tupaia longipes	Live trap	
39	Tupaiidae	Lesser treeshrew	Tupaia minor	Live trap	
40	Tupaiidae	Mountain treeshrew	Tupaia montana	Live trap	
41	Tupaiidae	Large treeshrew	Tupaia tana	Live trap	
42	Viverridae	Malay Civet	Viverra tangalunga	Camera trap	
43	Viverridae	Banded Linsang	Prionodon linsang	Camera trap	
44	Viverridae	Masked Palm Civet	Paguma larvata	Camera trap	

Alpha diversity

Both mountains showed a similar pattern for alpha diversity indices across elevations (Fig. 3; Table S2). Species richness and Shannon diversity were maximum in low elevations and decreased gradually toward high elevations. However, evenness was lowest at middle elevations (U-shaped) (Fig. 3). The high dominance of some species at middle elevations (e.g., mountain treeshrew) leads the Shannon diversity to sink at around 1,500 m in both mountains. However, Shannon diversity increases again toward the highest elevations due to the more even occurrence of the species in the small mammal communities at those elevations, despite lower species richness (Fig. 3). The species richness we report in low elevations for both mountains are above the upper 97.5% quantile of the expected richness expected under the hypothesis of the MDE (Fig. 3A).

Figure 3 Diversity indices across elevations.

Change of species richness (A), Shannon H’ (B), Simpson’s Diversity Index (C) and Pielou’s Evenness (D) across elevation for Mt. Kinabalu (circles) and Mt. Tambuyukon (triangles), with loess regressions (Mt. Kinabalu, dashed line; Mt. Tambuyukon, pointed line). Shaded areas in (A) represent the 2.5% and 97.5% percentiles of the species richness for expectations under the MDE (Mt. Kinabalu, brown; Mt. Tambuyukon, green), with closed symbols being the fitted values of species richness corrected for autocorrelation.

We detected a significant negative relationship between species richness and elevation (χ12 = 8.81, P = 0.003). However, species richness was not correlated with expected species richness under the MDE (χ12 = 0.54, P = 0.46) (Fig. 3).

Elevation was positively correlated with higher proportion of endemic species (χ² = 7.96, df = 1, P = 0.005) and a greater proportion of captures belonging to endemic species (χ² = 10.7, df = 1, P = 0.001) (Fig. 4).

Figure 4 Endemism with elevation.

Proportion of Bornean endemics (A) and proportion of catches belonging to Bornean endemics (B) across elevation. A confidence envelope for the observed values is represented as a shaded area from 1,000 bootstrap replicates.

Beta diversity

Variation in the species composition assemblages, or beta diversity, was very similar for both mountains (βSOR = 0.77 on Mt. Kinabalu and βSOR = 0.74 on Mt. Tambuyukon). Most of this beta diversity derived from the turnover component (βSIM = 0.73 for Mt. Kinabalu and 0.65 for Mt. Tambuyukon). The nestedness component was very low on both mountains (βNES = 0.04 for Mt. Kinabalu and 0.09 for Mt. Tambuyukon) (Fig. 5). This indicates that the assemblages at different elevations are not the product of species loss from the richest assemblages. Instead, they are singular assemblages with different species compositions. We found large dissimilarities in the turnover component between pairwise locations within each mountain (βsim) associated with < ~1,900 vs > ~ 1,900 m locations, compared to lower values within lowland or highland locations (Table S3). When removing endemic species from the dataset, βNES-end increased and βSIM-end decreased on both mountains (Fig. 5). The permutations indicated that this decrease in the turnover component (βSIM-end) was significant in Tambuyukon (βSIM-end = 0.36, P = 0.002), but not in Kinabalu (βSIM-end = 0.68, P = 0.36), and that the increase in nestedness (βNES-end) was significant for Mt. Tambuyukon (βNES-end = 0.38, P = 0.001), but only marginally significant for Mt. Kinabalu (βNES-end = 0.14, P = 0.05) (Fig. 5).

Figure 5 Beta diversity with and without endemics.

Sorensen dissimilarity (βSOR) decomposed into nestedness (βSNE) and turnover (βSIM) components, for Mt. Kinabalu and Mt. Tambuyukon (left/right depiction in each set of data points, respectively). Solid horizontal lines represent the observed values, while the dotted horizontal lines are the estimated beta diversity measures after removing endemic species from the matrix (βSIM-end, βSNE-end and βSOR-end). The random expectations for βSIM-end, βSNE-end and βSOR-end are represented from the dotted cloud with 5,000 permuted values (see main text), together with their corresponding 2.5% and 97.5% percentiles (vertical gray bar).

The clustering analysis grouped trapping locations in two main groups, above and below ~1,400 m (Fig. S3). Within the high elevation group, all locations above ~1,900 m clustered together.

Discussion

We report an extensive survey of non-volant small mammals within Kinabalu National Park, Borneo. A sampling effort of 8,231 trap-nights in Mt. Kinabalu (n = 2,044) and its neighbor peak Mt. Tambuyukon (n = 6,187) yielded a total of 512 individual records (Mt. Kinabalu, n = 213; Mt. Tambuyukon, n = 299) from 27 species (Mt. Kinabalu, n = 20; Mt. Tambuyukon, n = 23) (Table 1; Fig. 2). Records from camera traps and direct observations increased the total number of species recorded for Mt. Tambuyukon by 18 to a total of 44 (Table 3).

Mt. Kinabalu is a biodiversity hotspot for many taxa (Van Der Ent, 2013). Its mammal fauna has been studied for over a century (Oldfield, 1889; Whitehead, 1893; Emmons, 2000; Nor, 2001) and it is known to host 61 species of non-volant small mammals (Nor, 2001; Payne et al., 2007). We further explored for the first time the non-volant small mammal diversity along the complete elevational gradient in Mt. Tambuyukon (2,579 m), the third-highest peak in Borneo and only 18 km away from Mt. Kinabalu. Some important sightings on Mt. Tambuyukon included the orangutan (P. pygmaeus), which has an estimated population of only 50 individuals within Kinabalu Park boundaries (Ancrenaz et al., 2005). The Kinabalu ferret-badger (Melogale everetti) was a significant finding since it is the first official record of this species on Mt. Tambuyukon (Payne et al., 2007; Wilting et al., 2016). We trapped this species at 2,051 m on Mt. Tambuyukon and 3,336 m on Mt. Kinabalu. We identified a population of the summit rat (Rattus baluensis) on Mt. Tambuyukon, previously only known from Mt. Kinabalu. This species was common at high elevations and has its lower distribution limit at around 2,000 m. A population genetic analysis of the summit rats from Mt. Kinabalu and Mt. Tambuyukon demonstrated that they are currently genetically isolated (Camacho-Sanchez et al., 2018). We also make the first records of the mountain species Maxomys alticola, Hylomys suillus and Niviventer rapit on Mt. Tambuyukon.

Alpha diversity

Species richness peaked at low elevations on both mountains coinciding with the lowland dipterocarp forest. Then, it decreased gradually toward the highest elevations where it was lowest (Fig. 3). This pattern deviates from the expectations of the MDE (Fig. 3), and the MDE reported by Nor (2001) in a previous small mammal survey on Mt. Kinabalu with a very similar survey scheme to ours. Colwell & Lees (2000) suggested that a MDE should constitute the null hypothesis over which deviations should be interpreted. However, this point of view is not universal (Rahbek, 1995; McCain, 2007, 2009). Incomplete sampling of the low elevations happens regularly and can artificially create a MDE (Rahbek, 1995; Lomolino, 2001), which may explain the difference between Nor (2001) and our results for the same mountain. Our surveys detected 12 more species as compared to Nor (2001), including a climbing mouse, Chiropodomys pusillus, a tree rat Lenothrix canus, Prevost’s squirrel, Callosciurus prevostii, Jentink’s squirrel, Sundasciurus jentinki, the rats Maxomys rajah, Maxomys alticola and Sundamys muelleri, the non-native species Rattus exulans and Rattus tanezumi, two species of treeshrews Tupaia. longipes and T. minor, and two species of shrews, one trapped in a small Sherman trap, Crocidura sp., and one in a pitfall trap, Suncus sp.. The effects of an incomplete sampling should be more acute in lowland elevations where there is more habitat heterogeneity (Rosenzweig, 1992, 1995) and species might tend to occupy smaller ranges (Rosenzweig, 1995; Brown, 2014). However, we report the highest alpha diversity in the lowest elevations, which supports that our interpretations are not affected by this low-elevation sampling bias. Another observation supporting the comprehensiveness of our sampling is that we documented species in all sites between the lowest and highest occurrences, except for Maxomys whiteheadi on Mt. Kinabalu, and Maxomys ochraceiventer and N. rapit on Mt. Tambuyukon.

Conversely, the gradual decrease of species richness we recorded was explained by elevation alone. Even in the presence of a MDE, a gradual decrease of species richness with elevation seems to be a general pattern in mountain gradients (Rahbek, 1995). There are multiple factors that are correlated with elevation which have been proposed to explain diversity across mountains gradients, but disentangling their effects is difficult given the multicollinearity (Heaney, 2001; Fu et al., 2006; Kluge, Kessler & Dunn, 2006). The strong correlation of alpha diversity with elevation enables further discussion. Diversity has been proposed to peak with precipitation (Heaney, 2001), but on Mt. Kinabalu the peak of precipitation, at around 2,000 m (Kitayama, 1992), did not match the diversity peak. Productivity (Aiba, Takyu & Kitayama, 2005) and temperature (Kitayama, 1992) are negatively correlated with elevation in Kinabalu, and could potentially explain the diversity pattern. Nevertheless, it has been suggested that examining resource availability for this taxonomic group is more appropriate than simply looking at productivity (Brown, 2001; Heaney, 2001). This is a difficult variable to measure that was not incorporated in our original survey. A more plausible explanation for the change in diversity is area and habitat complexity. Available area (Camacho-Sanchez et al., 2018) and complexity of the forest (Kitayama, 1992) decrease with elevation on Mt. Kinabalu. The peak in diversity we find in low elevations is consistent with the spatial hypothesis which states that (1) at the regional level, larger areas (such as the lower elevations in mountains) have lower rates of extinction over speciation (Rosenzweig, 1992) and (2) that larger areas have more types of different habitats, so greater species diversity should be observed in larger areas (Rosenzweig, 1995). Perhaps, the relationship between area and diversity on elevational gradients along large mountains falls somewhere between these processes (McCain, 2007).

Beta diversity

The composition of the small mammal assemblage changed across elevation in a similar way for both mountains. Indeed, all trapping locations in high elevations (above ~1,900 m) were very similar in composition (Fig. S3). Mid-mountain locations showed intermediate compositions whereas the 500 and 900 m locations on both mountains also clustered together. This montane fauna transition was already identified by Nor (2001) to be at around 1,800 m, and it matches approximately the vegetation limit between the lower and upper montane forests on Mt. Kinabalu (Kitayama, 1992). The influence of this shift is reflected in the overall high turnover component of beta diversity (βSIM) for both mountains (Fig. 5). The pairwise turnover components (βsim) were highest between lowland—highland locations (Table S3). This indicates the lowland and highland communities are composed of different species, rather than the community with the lowest richness (highland) being a subset of the species present on the richest one (lowland). This pattern has already been described for other small mammals across several mountain systems (Mena & Vázquez-Domínguez, 2005). Unfortunately, there is no consistent data collection from other tall mountains in Sundaland to discuss a common mountain biogeography pattern in this region. Historical expeditions to Sumatra also point to a similar shift in vegetation structure and different mammal assemblages at high elevations above 2,000 m (Robinson & Kloss, 1918, 1919; Miller, 1942).

Pattern of endemism

Beyond the net alpha and beta diversity we describe, we put special attention on the nature of the species that could be driving these diversity patterns. Our models predicted that the proportion of Bornean endemism as well as the proportion captures belonging to endemic species increased with elevation (Fig. 4). At the same time, we found that the high-elevation endemics were responsible for the high turnover component in beta diversity on Mt. Tambuyukon (Fig. 5). The species that had the greatest contribution to this high-elevation endemism were Rattus baluensis, T. montana, Sundasciurus everetti and Maxomys alticola (Table S1). These species are restricted to mountain areas in northern or central-northern Borneo, and were present in high abundance and evenness from 2,000 m (Table S1). Nor (2001) reported high trapping success on Mt. Kinabalu, associated with higher abundance at high elevations, but he did not record Maxomys alticola. Heaney (2001) also recorded the highest abundances in the top elevations in the Philippines, and a peak of endemics at higher elevations. Three additional trapped species are mountain endemics (Melogale everetti, Sundasciurus jentinki and N. rapit) but contributed less to our analysis because of their lower densities or detectability. This pattern of high-elevation endemics could be even more pronounced as the distribution and taxonomy of the highland mammals also found on other islands in Sundaland are further updated and revised (i.e., Hylomys suillus and Sundamys infraluteus; Camacho Sánchez, 2017). A majority of the lowland species are widespread, and also distributed across other Sundaland landmasses such as Sumatra and the Malay Peninsula (Corbet & Hill, 1992).

The pattern of mountain endemics on Mt. Kinabalu has previously been described for other taxa (Merckx et al., 2015). For mammals, this has also been observed by Heaney (2001), Sánchez-Cordero (2001), Swenson et al. (2012) and Chen et al. (2017) on other mountains. Mt Kinabalu is unique in Sunland due to its high elevation (4,095 m) and the scarcity of nearby peaks above 2,000 m. Higher isolation on mountain peaks boosts mountain endemism worldwide (Steinbauer et al., 2016), which could explain the greater endemism in the higher elevations in the Kinabalu range. For instance, a pattern of mountain endemism linked to divergence in allopatry induced by isolation from the combination of topography with past climate changes has been described in Bornean birds (Sheldon, Lim & Moyle, 2015; Moyle et al., 2017; Manthey et al., 2017). A similar pattern of high mountain endemism driven by intra-island speciation has been reported for shrews in Sumatra and Java (Esselstyn et al., 2013; Demos et al., 2016). Presumably, the high degree of isolation of the habitats on the higher slopes of Mt. Kinabalu help to maintain a highly endemic community. This could be due to reduced dispersal and colonization to/from nearby similar areas as proposed by Steinbauer et al. (2016).

Conclusions

We found a decline in small mammal diversity from low to high elevations on both Mt. Kinabalu and Mt. Tambuyukon. This pattern differs from the MDE previously described for Mt. Kinabalu and other mountains worldwide. The decrease in diversity with elevation is concordant with the spatial hypothesis predicting higher diversity in lowlands driven by historically larger areas with less extinction and more habitat heterogeneity. However, we cannot exclude other climatic or ecological hypothesis. Endemic species were in higher proportion and more abundant in higher elevations and they drove the turnover component of beta diversity. The high number of mountain endemics point to historical factors as important drivers of the biogeography in this region.

Supplemental Information

Supplemental Information 1 Raw data of animals sampled and trapping effort.

Click here for additional data file.

Supplemental Information 2 Species accumulation curves across elevations in Mt. Kinabalu (left) and Mt. Tambuyukon (right).

Click here for additional data file.

Supplemental Information 3 Pictures from camera traps.

Click here for additional data file.

Supplemental Information 4 Cluster analysis using neighbor joining from Sorensen distances.

Click here for additional data file.

Supplemental Information 5 Summary table of species per elevation.

Small mammals trapped during field surveys. Mt. Tambuyukon was surveyed at 500, 900, 1,300, 1,600, 2,000 and 2,400 m. Mt. Kinabalu was surveyed at 500, 900, 1,500, 2,200, 2,700, 3,200 m. Columns are headed with the elevation (m), where the same (or similar) elevation was sampled between the two mountains (Mt. Tambuyukon/Mt. Kinabalu).

Click here for additional data file.

Supplemental Information 6 Diversity indexes.

Diversity calculations for both mountains, across elevations (H′, Shannon diversity index; D, Simpson diversity index; S, species richness; J′, Pielou’s evenness index).

Click here for additional data file.

Supplemental Information 7 Pairwise dissimilarity between trapping locations.

Pairwise dissimilarity between trapping locations based on Sorensen index (βsor), and its decomposition into the turnover (βsim) and nestedness (β sne) components.

Click here for additional data file.

This research would not have been possible without the help and assistance from many individuals. This includes logistical support from Maklarin Lakim, Alim Buin, Paul Imbun, Justin Sator, Robert Stuebing, Fred Sheldon and Konstans Wells. Kristofer Helgen assisted in many aspects of this work. Darrin Lunde is thanked for assistance with preparation prior to the field expeditions, and for the use of traps from the Division of Mammals at the National Museum of Natural History. Larry Rockwood allowed use of several of the camera traps, and for that we are grateful. Megan Whatton assisted with the calculations for the camera trap data. Jeffrey Hanson advised on how to correct for autocorrelation. Rebecca Rowe provided valuable insights for the review process. Field Crew: Rose Ragai, Lyndon Hawkins, Flavia Porto, Manolo Lopez, Paco Carro, Ipe, and Anzlys. Logistical support was provided by the infrastructures offered by Doñana’s Singular Scientific-Technical Infrastructure (ICTS-EBD).

Additional Information and Declarations

Competing Interests

Author Contributions

Animal Ethics

Field Study Permissions

Data Availability

Fred Tuh Yit Yuh is an employee of Sabah Parks.

Miguel Camacho-Sanchez conceived and designed the experiments, performed the experiments, analyzed the data, prepared figures and/or tables, authored or reviewed drafts of the paper, approved the final draft.

Melissa T.R. Hawkins conceived and designed the experiments, performed the experiments, analyzed the data, prepared figures and/or tables, authored or reviewed drafts of the paper, approved the final draft.

Fred Tuh Yit Yu conceived and designed the experiments, authored or reviewed drafts of the paper, approved the final draft.

Jesus E. Maldonado conceived and designed the experiments, authored or reviewed drafts of the paper, approved the final draft.

Jennifer A. Leonard conceived and designed the experiments, authored or reviewed drafts of the paper, approved the final draft.

The following information was supplied relating to ethical approvals (i.e., approving body and any reference numbers):

Animal care and use committees approved the protocols: Smithsonian Institution, National Museum of Natural History, Proposal Number 2012-04 and Estación Biológica de Doñana Proposal Number CGL2010-21524.

The following information was supplied relating to field study approvals (i.e., approving body and any reference numbers):

Field research was approved by the Economic Planning Unit (Malaysia) (100-24/1/299), the Sabah Biodiversity Council (JKM/MBS.1000-2/2(104)) and Sabah Parks (TS/PTD/5/4 Jld. 45 (33) and TS/PTD/5/4 Jld. 47 (25)).

The following information was supplied regarding data availability:

The raw data is available in the Supplemental Files and Zenodo: Miguel Camacho. (2019, August 29). csmiguel/smallmammals_Kinabalu: Revision August 2019 (Version 0.2.0). Zenodo. DOI 10.5281/zenodo.3381149.

All analyses and figures are reproducible.

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
