# Peer review of "Endemism and diversity of small mammals along two neighboring Bornean mountains"

_PeerJ, doi:10.7717/peerj.7858_

## Round 0.1 · original submission · Major Revisions

This paper describes an extensive mammal trapping and collection effort on two mountain peaks in Borneo. It presents an impressive dataset and describes the patterns of diversity observed among the small mammal community. Although the reviewers expressed concerns with the applications of this data to test theories of elevational biodiversity patterns, I believe the paper is strong based on the description of the fieldwork and the documentation of natural history patterns. PeerJ does not judge papers on the basis of "impact, degree of advance, novelty" etc. (see https://peerj.com/about/editorial-criteria/ ).

My recommendation is to address the following in the revision:
1. Include sufficient introduction and background to demonstrate how the work fits into the broader field of knowledge. Information from line 296+ on the MDE could be moved to the introduction. Besides testing the MDE and other patterns, the work has value in simply documenting the biodiversity of the community. This is mentioned in Line 292+ but perhaps you could discuss the threats facing mammals in this region.
2. Define the research question, which must be relevant and meaningful. What is the knowledge gap and how does your study contribute to filling that gap?
3. Based on Fig 2, I can visually see why you do not support the MDE, but is there any null-modeling or other test that can confirm this conclusion? How do other researchers test this?
4. Address the question of autocorrelation in the data (Reviewer 1)
5. Address Reviewer 2's comments on the Methods and Results sections. Ensure that the questions about the timing of field seasons on each mountain, and why the arboreal and pitfall trap data were not included, are answered.
6. Reduce the number of figures and tables to reduce complexity of the paper and focus on what is the main point pf the paper. You can drop the discussion Bray-Curtis dissimilarity indices and Fig 6. You can also drop Fig 5; I do not see it as critical to the message of the paper.

The supplementary data file is great and will likely be used by many other researchers!

Detailed comments:
Abstract: First sentence is not clear and does not introduce the purpose of the paper. Maybe state the project goals (from Line 52) in the abstract.
Line 174: The first sentence on trapping results is obvious. Make a more meaningful statement here.
Line 253: What are the implications of the results for beta diversity? Need to justify why you report this at all.
Line 330: The word "cohesive" is not appropriate here. I believe you want "consistent".
Line 348: "constraints" should be "constraints"
Throughout: use "elevation" not "altitude".
Table 2: separate out the two mountains. The '/' is confusing. Also need to rewrite the figure caption to be clearer.
Table 3: Which mountain were the camera traps on? Are these locations marked in Fig 1?
Table 4: Why is this list only for Mt. Tambuyukon? Please provide the list for the other mountain as well.
Fig 2: Label the y-axis of each panel with the corresponding diversity index. Place the legend either at the top, bottom, or to the side of the figure panels.
Fig 3: What methods were used to calculate the species accumulation curves? Is this documented in the methods section? I would like to know whether these curves asymptote or not.
Fig 4: This figure is too complex. Remove the accumulated trapping effort for each mountain. It is depicted in Fig 3, which is very nice. Maybe indicate the Bornean endemics with an asterisk on the name, since you are removing Fig 5.

Reviewer 1 ·

Basic reporting

All points are integrated into one document; see General comments for the author

Experimental design

All points are integrated into one document; see General comments for the author

Validity of the findings

All points are integrated into one document; see General comments for the author

Additional comments

The manuscript “Small mammal diversity along two neighboring Bornean mountains”, by Melissa T. R. Hawkins and coauthors aims to describe distribution patterns of small mammals along elevation gradients on two mountains in the Kinabalu National Park. They perform a more intensive sampling than a previous work in order to evaluate if they would still detect a Mid-domain effect. The manuscript is well written in terms of English language, they follow adequate ethical and sampling protocols, perform a well designed sampling work and methods are described with enough detail to replicate.

Main comments
However, I am concerned with different aspects of the research that I explain as follows:

Introduction: the authors start the introduction describing what they did “We surveyed the diversity of the non-volant small mammals along the elevational gradients of
two neighboring mountains on the island … “ and what had already been done “Previous surveys of elevational transects have measured diversity”, with a rather brief comment on the Mid-Domain Effect (lines 31-51). Hence, it lacks a sound description of the theoretical background that is needed to put the study into context and present clear scientific questions.

As a result, I consider the authors work with a research question that falls very short of the potential of their data; namely their objectives [(1) to better characterize the diversity of the small mammal community along elevational gradients in Kinabalu National Park, (2) to evaluate whether the MDE previously described is robust to incomplete sampling of the low-elevations, (3) to determine if the same pattern of diversity across the altitudinal transect is the same on the two mountains], do not pose a novel, significant research question. As stated, they do perform a better description of the small mammal diversity, based on a more adequate sampling, of a elevational pattern that was already known.
Also, they do not justify why to compare the two mountains, what is the scientific question behind it. As a consequence, the information and data they obtain do not allow for a strict and thorough discussion of the evolutionary and historical factors that could have driven small mammal diversity with elevation in this context (their objective 4).

It is important to highlight that for the very basic question of the Mid-domain effect (MDE), one needs to determine if the data being analyzed (in this case, diversity along the elevation gradient) is autocorrelated. Autocorrelation is commonly present in this kind of data, thus analyses need to be done taking that into consideration. There is plenty of literature about Mantel tests, autocorrelation, variograms, etc., but see the published work of Christy MaCain for specific examples regarding MDE, Pierre Legendre for autocorrelation. Indeed, results can be completely different once the correction for autocorrelation has been performed.

I recommend that the authors think of more complex and novel questions that they can evaluate with the ample information they have gathered. Published literature about species richness, beta diversity and the sort, for diverse gradients including elevational, latitudinal, etc. are abundant. I am sure the authors can perform a more novel and gap filling study.

Reviewer 2 ·

Basic reporting

The manuscript is poorly written. I believe the authors must rewrite the entire manuscript.

Experimental design

I found the general idea of looking for diversity patterns interesting.

Validity of the findings

The manuscript is really hard to follow due to the poorly written style of the authors.

Additional comments

The authors presented an observational study aimed to look at patterns of small mammals diversity along two elevational gradients in Borneo. I think that the general topic addressed in this manuscript is interesting, the methods used are straightforward and the amount of work in the field is to say at least impressive. A major but maybe fixable shortcoming of this manuscript is the writing style. I just listed the major issues that I see in each section of the manuscript.
(1) The Abstract is hard to follow. I read the first sentence several times and I still do not understand what the authors tried to say.
(2) The Introduction’s first sentence is what the authors actually did in the field. The authors should begin the introduction with a very general overview of the topic. Then, you can narrow the topic until you reach the aim of the study and the hypotheses tested.
(3) Methods: the Study site section is way too long with too much detail. Field survey, did the authors conduct the surveys in two consecutive field season in both mountains, or they only conducted two field seasons in Mt. Tambuyukon? Why the surveys were conducting in different time of the year? What is the rationality behind this change?
(4) Results: Line 161 – “The trap success calculations were done excluding pitfall traps and arboreal traps (due to inconsistent placement of traps).” This problem only happened in Mt. Tambuyukon? I do not understand what happen with the pitafall traps and arboreal traps. 
The results are hard to understand because of these distracting issues.
Except for the Diversity subheading the whole Results section must be rewritten. I strongly believe that the natural history is really important, and looks like the authors know a lot about it, but it is hard to pick the main results among all the details presented by the authors. Additionally, there are too many tables and figures.
(5) The Discussion section presents the same problems that the rest of the manuscript, except for the subheading Conclusions. The first paragraph of the section should be a brief summary of the most important results.

If the authors are willing to re-write the entire manuscript, the study would be suitable for publication in PeerJ. However, a thorough revision of the manuscript should be conducted once the writing problems are fixed. I am sorry I cannot be more positive, especially because both the interest and importance of this work are clear, but I hope these comments can help the authors.

---

## Round 0.2 · Major Revisions

The authors have taken several points into account in their revisions, but several problems still remain.

First and foremost, the framework of the study needs to be clarified further and objectives and predictions made clearer.

The text also need to be thoroughly reviewed to remove unnecessary, repetitive sentences and speculative ideas, as it is still difficult to read in its current form.

Finally, the problem of autocorrelation still remains and needs to be dealt with appropriately (by correcting for it in the analyses). The reviewer has made a useful suggestion regarding how to achieve this.

Reviewer 1 ·

Basic reporting

All points are integrated into one document; see General comments for the author

Experimental design

All points are integrated into one document; see General comments for the author

Validity of the findings

All points are integrated into one document; see General comments for the author

Additional comments

I consider the authors made significant improvements following the reviewers and editor suggestions and considerations, rendering a manuscript with the potential to be published in PeerJ.
However, there are still some important problems that need to be addressed and corrected.


General comments:

The writing of ideas still needs to be clearer, direct, avoiding loose, out-of context phrases
Avoid de use of “very”, it is subjective and non-scientific (lines 36, 42, and all throughout the manuscript).

Specific comments:

Abstract and Introduction
Line 33 : delete second hypothesis

Lines 35-37 The entire phrase “These data enable … patterns” is confusing; what is meant by long elevational gradients? similar mountains –are very similar? What selective drivers in the environment determine what patterns? Speculative and unclear

Lines 43-46: this phrase is out of context here .. eliminate and concentrate on the biodiversity patterns background.

Lines 71-75 : another confusing phrase with several unconnected ideas ¿The other side? A simpler writing is: “Knowledge about tropical biodiversity is still incomplete”. … ¿More boots on the ground … in (on) the news … totally out of context.

Lines 97-102: the manuscript still lacks a clearer objective and specific predictions. Namely, what of the different “standing” hypotheses were considered? what underplaying assumptions, some, all? What would the authors expect to find given the nature of the mountains, and the hypotheses and assumptions? You have these in the discussion (lines 310-314)

Methods
The problem with autocorrelation has not been solved.
First, why estimating a Mantel test between elevation and similarity? the mid-domain effect refers to species richness (or Beta diversity) within a mountain, and then comparison between mountains can be done.
A Mantel is only a part of the needed analyses. The authors indicate in their answer that “However, our overall goal is to look at the curve of diversity with elevation, autocorrelations effects included. We are not testing the mechanisms causing the diversity pattern. Therefore, in our particular case, as we are considering only one variable, elevation, as our only explanatory variable for diversity, we think our approach is correct”
Autocorrelations effects included does not make sense. You do need to correct for autocorrelation,
It does not matter that only elevation is being considered, you are testing for a Mid-Domain effect hence the relationship has to be corrected for autocorrelation (see Mena & Vázquez-Domínguez. 2005 Global Ecology and Biogeography, 14(6): 539-547, for a detailed explanation)

Discussion
I urge the authors to work some more on the discussion, making an effort to summarize ideas, shortening and eliminating repetitive or speculative statements (in particular Pattern predictability and Conservation implications). This will strengthen the work and make it more succinct and fluent.

Lines 327-329 : see precisely Mena & Vázquez-Domínguez 2005.

---

## Round 0.3 · Minor Revisions

We have received two reviews for your revision. In general, the authors are commended for their extensive revisions. As a result, one of the previous reviewers (R1) was generally satisfied with revisions performed. An additional ‘new’ reviewer provided a few key points that will further improve the quality of your manuscript and in particular the hypotheses and framework of the study.

Reviewer 1 ·

Basic reporting

See the "General comment for the author" below

Experimental design

See the "General comment for the author" below

Validity of the findings

See the "General comment for the author" below

Additional comments

The authors took into consideration all of my (and other reviewers) comments and fully transformed the manuscript; they incorporated the new analyses required, rewrote some of the Introduction to incorporate the new objectives/questions, and also the Discussion, which changed accordingly with their new results.
Hence, I have no further comments and consider the manuscript is now appropriate for publication in PeerJ.

Reviewer 3 ·

Basic reporting

This manuscript represents an impressive body of work on Borneo's small, non-volant mammals. In the current state, however, major changes need to be made. First, the English reads quite strangely, especially in the introduction. I have made a few specific comments regarding this in my line-by-line comments. As a whole the manuscript needs language attention. A second issue is the discrepancy between hypotheses and statistical methods. In the introduction the only hypothesis that was stated was that mountain endemics shape diversity, versus the null that species are distributed everywhere. The authors appear to be testing hypotheses about the mid-domain effect, and should clearly state so in the introduction.

Experimental design

In the current state the research question is not well defined. A more comprehensive set of hypotheses need to be outlined.
The statistical methods were sound, though a few more details need to be provided.

Validity of the findings

The statistical methods appear to be solid, though I have a few comments in the line-by-line. The most challenging finding in the manuscript is the species-level identifications made using live-captured animals and a field guide with hand-drawn illustrations. Though I am not an expert on some of these species, I know, according to many experts, that many of these murine rodent and crocidurine shrew species can be incredibly hard to identify based on external characters alone. Many researchers would argue that DNA and/or cranial characters are critical in identifying these species. This is speculation, and I do not mean to diminish the authors expertise, however I find the consistent and accurate identification of 5 Maxomys species, a few of which are very poorly know, a bit suspect.

As pointed out in the conclusions, their survey results—which found no support for the MDE—differed from past survey results that found support for the MDE. This is to be expected. A single survey can never capture the full community of a place as diverse and challenging as Kinabalu. It may make more sense to combine modern and historic survey efforts to test these large-scale hypotheses.

Additional comments

The effort put into collecting these data is very impressive! The manuscript tests some interesting hypotheses about patterns of species richness along elevational gradients, but the hypotheses and framework remain unclear. I recommend working on simple statements of predictions in the introduction, and focusing on how these relate to the methods and results in those respective sections.

Annotated reviews are not available for download in order to protect the identity of reviewers who chose to remain anonymous.

---

## Round 0.4 · accepted · Accept

I am happy with the final revisions made on the manuscript. All additional points have been satisfactorily addressed in this revised version.